# Characterization of Soil Bacterial Communities in Different Vegetation Types on the Lava Plateau of Jingpo Lake

**DOI:** 10.3390/microorganisms13071648

**Published:** 2025-07-11

**Authors:** Yanli Zhang, Jiaxing Huang, Jiaxin Xue, Kaining Zhang, Xintong Chen, Jianhui Jia, Qingyang Huang

**Affiliations:** 1School of Life Science and Technology, Mudanjiang Normal University, Mudanjiang 157011, China; swxzyl@126.com (Y.Z.); hjx19845715212@163.com (J.H.); m18745013336@163.com (J.X.); 18234626459@163.com (K.Z.); 19904538315@163.com (X.C.); swxjjh@126.com (J.J.); 2Institute of Natural Resources and Ecology, Heilongjiang Academy of Sciences, Harbin 150040, China

**Keywords:** lava platform, soil physical and chemical properties, rhizosphere bacterial community, PICRUSt2 functional prediction

## Abstract

To explore the interactions within the vegetation–soil–microorganism continuum on the Jingpo Lake lava platform, five vegetation types—grassland (GL), shrubland (SL), deciduous broad-leaved forest (DB), coniferous and broad-leaved mixed forest (CB), and coniferous forest (CF)—were examined. Significant differences in the soil physical and chemical properties were identified among these types (*p* < 0.05). The soil bacterial community structures also varied significantly (*p* < 0.05), with Actinobacteriota, Proteobacteria, and Acidobacteria as the dominant phyla, exhibiting notable genus-level differences (*p* < 0.05). The soil organic matter (SOM), available nitrogen (AN), total nitrogen (TN), and soil water content (SWC) were significantly correlated with the bacterial community structure (*p* < 0.05 or *p* < 0.01), acting as key determinants of the microbial community structure and function. PICRUSt2 functional predictions revealed significant variations in the metabolic functions of the soil bacterial communities across vegetation types, indicating distinct functional specializations. In conclusion, the Jingpo Lake lava plateau harbors abundant bacterial resources. When devising vegetation adaptation strategies, it is essential to take into account variations in the rhizosphere soil bacteria across different vegetation types. Furthermore, prioritizing the implementation of forest vegetation is crucial in the adaptive management of the lava plateau. This approach holds significant implications for studying the bacterial diversity in the lava plateau and exploring the cultivation and application of functional bacteria in extreme environments.

## 1. Introduction

The lava plateau ecosystem is a complex and unique system that plays a pivotal role in global ecology, contributing significantly to biodiversity and ecological balance. It plays a crucial role in the global carbon cycle [1,2], provides insights into biological evolution and ecological adaptation [3], assists in addressing climate change [4], supports geological research [5], and enables the study of co-evolution between volcanic activity and terrestrial biomes [6]. Additionally, it creates favorable conditions for the flourishing of the Mesozoic terrestrial ecosystem [7]. Influenced by natural soil-forming factors and human activities, the nutrient content of lava plateau soils exhibits distinct temporal and spatial patterns, impacting regional vegetation distribution. The vegetation on these plateaus can be classified into five kinds based on the integrity of the succession sequences and the functional driving factors: grassland, shrubland, deciduous broad-leaved forest, coniferous and broad-leaved mixed forest, and coniferous forest [8].

Soil microorganisms are crucial for decomposing animal and plant residues and for driving ecosystem energy cycles and material flow. Bacteria, as a dominant group among soil microorganisms, significantly contribute to nutrient cycling and organic matter decomposition in terrestrial ecosystems [9,10]. They enhance plant growth and facilitate plant community succession by ameliorating plant–soil negative feedback. The soil bacterial community is highly responsive to environmental changes influenced by plants, which alter the soil conditions (e.g., temperature, moisture, pH, carbon, nitrogen, phosphorus) through litter and root exudates. Approximately 30% of plant photosynthetic products are released into the soil via roots, impacting the structure, function, and diversity of soil microorganisms based on the distinct biochemical characteristics of plant litter and root exudates [11,12]. The kind of vegetation mostly influences the metabolic activity, the carbon utilization rate, and the microbial biomass of the soil microbial community. As the soil microbial community experiences succession, the population of soil microorganisms and the rate of carbon use progressively rise [12,13]. Furthermore, the vegetation quantity and the chemical composition are key factors influencing the soil microorganism diversity. Soil microorganisms actively participate in biochemical processes and organic matter transformations, regulating the soil’s nutrients and physical and chemical properties and indirectly influencing plant growth and community succession. Consequently, a closely intertwined system involving soil microorganisms, vegetation, and soil quality evolves [14]. The current research on soil microbial community characteristics has primarily concentrated on loess hills, alpine grasslands, and subtropical mountains, with limited investigations on the interplay between the soil bacterial community structure and vegetation in volcanic lava platforms, which are characterized by shallow ecological environments and exposed rocks [7,15].

The Jingbo Lake World Geopark, situated in the southeastern region of Heilongjiang Province, showcases volcanic groups that erupted multiple times between 520 and 550 years ago, leading to the destruction of the original vegetation and soil [16,17]. This event gave rise to a distinctive, complete, and scarce volcanic geological landscape, representing a valuable natural heritage resulting from Earth’s evolutionary processes. The volcanic lava platform at Jingpo Lake serves as a natural laboratory for investigating various significant scientific inquiries, such as the flora of volcanic ecosystems, soil nutrient dynamics, and soil microbial communities. While the existing literature extensively covers soil bacterial communities in terrestrial ecosystems, limited information is available regarding lava platforms, particularly concerning comparative analyses of the soil microbial community characteristics across different vegetation types on these platforms, thus warranting further investigation [16,18]. The present study investigated the primary succession process on the Jingpo Lake lava platform by comparing five vegetation types: grassland, shrubland, deciduous broad-leaved forest, coniferous and broad-leaved mixed forest, and coniferous forest. This study aimed to (1) evaluate the physical and chemical properties of soils across different vegetation types; (2) assess the structure and diversity of the soil bacterial communities; (3) identify functional groups within the soil bacterial communities; and (4) investigate the factors influencing the soil bacterial community characteristics. By achieving these objectives, this study aimed to deepen our understanding of the environmental impact of lava plateaus on the soil, contribute to theories concerning soil microorganisms in lava plateau ecosystems, and lay the scientific groundwork for global vegetation restoration mechanisms and sustainable development in unique ecosystems.

## 2. Materials and Methods

### 2.1. Site Description and Soil Sampling

The Jingpo Lake lava platform, situated between 128°30′–129°10′ E and 44°′–44°20′ N (Figure 1), is characterized by a “table-like” topography with a flat surface and steep edges, resulting from an “inverted terrain.” This region exhibits a continental monsoon climate characteristic of the moderate temperate zone, distinguished by prolonged, frigid winters and brief, cool summers. The yearly mean temperature is 4.3 °C, accompanied by an average relative humidity of 71.5% and an annual precipitation of 619.8 mm [19,20]. The predominant soil types include basic rock volcanic lithic soil and dark brown volcanic ash soil, supporting a variety of vegetation such as lichen moss, grassland, shrubland, deciduous broad-leaved forest, coniferous and broad-leaved mixed forest, and coniferous forest [8].

### 2.2. Soil Collection and Treatment

In July 2023, sampling sites were established across five vegetation types: grassland, shrubland, deciduous broad-leaved forest, coniferous and broad-leaved mixed forest, and coniferous forest, as depicted in Figure 1. Each plot included a 10 m × 10 m quadrat, within which five 1 m × 1 m sub-quadrats were positioned at the quadrat’s corners and center, following the X-sampling method for a community analysis. The dominant species were identified based on the plant occurrence frequency, coverage area, and relative biomass (Table 1). The predominant soil types included basic rock volcanic lithic soil and dark brown volcanic ash soil, supporting a variety of vegetation such as lichen moss, grassland, shrubland, deciduous broad-leaved forest, coniferous and broad-leaved mixed forest, and coniferous forest [2].

### 2.3. Determination of Soil Physico-Chemical Properties

The soil bulk density (BD) was measured using the ring knife weighing technique, while the soil water content (SWC) was evaluated by the drying method. The soil pH was assessed with a pH meter, and the electrical conductivity (EC) was evaluated using a conductivity meter. The soil organic matter (SOM) was measured using the external heating method with potassium dichromate. The total nitrogen (TN) was quantified via the Kjeldahl method, whereas the total phosphorus (TP) was evaluated using the HClO_4_-H_2_SO_4_ digestion technique. The total potassium (TK) was measured using NaOH melting flame photometry. Alkaline dissolved nitrogen (AN) was quantified using the alkaline diffusion technique, whereas quick-acting phosphorus (AP) was assessed via the 0.5 mol/L alkaline dissolution diffusion method. The available potassium (AK) was measured utilizing the 1 mol/L NH_4_OAc-0.01 mol/L EDTA extraction technique [21,22,23].

### 2.4. High-Throughput Sequencing of Soil Bacteria

Microbiome DNA was extracted with the MoBio PowerSoil^®^ DNA Isolation Kit (product No. 12888-100, MO BIO Laboratories, Carlsbad, CA, USA). The DNA quality was assessed via agarose gel electrophoresis, and quantification was performed using an ultraviolet spectrophotometer. The V3-V4 region of the 16S rDNA gene was PCR-amplified with the primers 341F (5′-CCTACGGGGNGGCWGCAG-3′) and 805R (5′-GACTACHVGGGTATCTAATCC-3′). The PCR products were purified using AMPure XP beads (Beckman Coulter Genomics, Danvers, MA, USA) to eliminate primers and non-specific fragments. The purified amplicons were quantified with an Agilent 210 Bioanalyzer (Agilent, Santa Clara, CA, USA) and Illumina library quantification kits (Kapa Biosciences, Wilmington, MA, USA). Sequencing was conducted on the Illumina NovaSeq 600 (PE250) platform by Hangzhou LC-Bio Technology Co., Ltd. (Hangzhou, China). The read length was 2 × 250 bp, yielding an average of 80,000 reads per sample post-quality filtering, ensuring an adequate sequencing depth. Negative controls were included throughout DNA extraction and sequencing to monitor contamination [24].

The raw sequences underwent processing via the QIIME2 pipeline (version 2023.2). Cutadapt was employed to remove primers and low-quality sequences. DADA2 was used for denoising and clustering into operational taxonomic units (OTUs) with the following quality-filtering thresholds: a maximum expected error of 2 and truncation lengths of 240 bp for forward reads and 200 bp for reverse reads. The taxonomic classification utilized the SILVA database (release 138) at a 99% similarity threshold. The OTU table was normalized using DESeq2 (CSS method) for a subsequent statistical analysis and visualized in Python 3.13.5. The raw sequencing data were deposited in the NCBI Sequence Read Archive (SRA) under the accession number [PRJNA1285586].

The functional potential of bacteria was forecasted utilizing PICRUSt2 (Phylogenetic Investigation of Communities by Reconstruction of Unobserved States, version 2). The operational taxonomic units (OTUs) were aligned using the Greengenes reference phylogenetic tree (version 13.5). Anticipated gene families were aligned with KEGG Orthologs (KOs) and structured into hierarchical KEGG pathways (Levels 1–3), utilizing a nearest sequenced taxon index (NSTI) threshold of less than 2.0.

Functional annotations were derived from the KEGG database (Release 107.0), consistent with microbial metabolic pathway classifications. This approach enabled high-resolution functional profiling when metagenomic sequencing was unavailable.

### 2.5. Data Processing and Analysis

The analysis employed a one-way analysis of variance (ANOVA) test and a chi-square test, utilizing the SPSS 27.0 statistical analysis software to assess the impact of various vegetation types on the soil physico-chemical properties. The data were then plotted using the Origin 2021 software. A redundancy analysis (RDA) was conducted and visualized in March 2025 on the Lianchuan BioCloud platform (https://www.omicstudio.cn/). In the analysis of differential RNA expression, negative binomial distribution tests were performed using DESeq2 or edgeR to calculate the *p*-values of the gene expression differences, and multiple hypothesis tests were corrected by the Benjamini–Hochberg method (FDR < 0.05). The significance threshold was set at *p* < 0.05 to ensure the statistical reliability of the results. The RDA analysis evaluated the significance of environmental factors on the bacterial community variation through 999 permutation tests, and an FDR (false discovery rate) correction was applied to multiple comparisons. The significance threshold was set to *p* < 0.05 [25].

## 3. Results and Analyses

### 3.1. Soil Physico-Chemical Properties of Different Vegetation Types

Marked discrepancies in the soil physical and chemical characteristics were observed among different plant types (Table 2). The BD was the highest in the SL, with the BD values in the SL and GL significantly surpassing those in the forest types, including the CF, CB, and DB (*p* < 0.05). This tendency was contrary to that of the SWC. The soil pH for all vegetation species was somewhat acidic, varying from 6.22 to 6.74, with a statistically significant difference between the GL and CF (*p* < 0.05). The EC displayed substantial variation among the vegetation types (*p* < 0.05); the CF demonstrated the highest EC, at 133.84 μs·cm^−1^, while the lowest was recorded for the SL, at 95.57 μs·cm^−1^. The trends for the SOM, TN, TP, and TK were uniform, adhering to the sequence of CF > CB > DB > SL > GL. Comparable patterns were noted for the AN, AP, and AK among the vegetation types (Table 2).

### 3.2. Soil Bacterial Community Composition in Different Vegetation Types

The high-throughput sequencing of the 16S genes from soil bacteria across the five vegetation types produced 1,032,426 effective sequences. At the phylum level (Figure 2a,b), 53 phyla were identified, with 9 phyla exceeding a 1% relative abundance. Actinobacteria, Proteobacteria, and Acidobacteria dominated, comprising 64.0% of the bacterial community in the Jingpo Lake lava platform. Significant differences were observed among the soil bacterial communities across vegetation types. The relative abundance trends of the phyla were similar in the GL and SL, whereas in the CF, CB, and DB, Proteobacteria were significantly more abundant than the other phyla (*p* < 0.05). Actinobacteria showed a significantly lower relative abundance in the forest types compared to the GL and SL (*p* < 0.05), contrasting with Acidobacteria. Planctomycetota were significantly more abundant in the DB than in the other types (*p* < 0.05).

At the bacterial genus level (Figure 2c,d), 26 genera exhibited a relative abundance exceeding 1%. A bar chart was constructed to depict the top 10 genera by relative abundance. *Actinobacteriota_unclassified* predominated in the GL and SL, *Rokubacteriales_unclassified* predominated in the DB, and *Candidatus-Udaeobacter* predominated in the CF and CB. *Candidatus-Udaeobacter*, *RB41*, *KD4-96—unclassified*, and *AlphaProteobacteria_unclassified* were significantly more abundant in the CF compared to the other vegetation types (*p* < 0.05). *Actinobacteriota_unclassified*, *Gaiellales_unclassified*, and *Gemmatimonadaceae_unclassified* were significantly more prevalent in the SF and GL (*p* < 0.05). In the GL, *MB-A2-108_unclassified* was notably more abundant than in the other vegetation types, contrasting with the trend for *Vicinamibacterales_unclassified*. *Rokubacteriales_unclassified* was significantly more abundant in the DB and CF than in the other three vegetation types (*p* < 0.05).

### 3.3. Soil Bacterial Diversity in Different Vegetation Types

The coverage indices for all five vegetation types exceeded 97%, signifying that the sequencing outcomes of the samples accurately represent the bacterial community data of the test samples. The observed number of OTUs, the Shannon index, and the Chao1 index of soil bacteria across the five vegetation types exhibited significant differences, with the CF sample plot demonstrating markedly higher values in all three metrics compared to the other four vegetation types (*p* < 0.05). The observed OTUs and the Chao1 index in the SL were considerably lower than those in the other four vegetation types (*p* < 0.05). The Pielou-e index ranged from 0.92 to 0.93, indicating an even distribution and balance of the bacterial flora (Table 3).

### 3.4. Prediction of Soil Bacterial Community Function in Different Vegetation Types

Using PICRUSt2, we predicted the soil bacterial functions across various vegetation types on the Jingpo Lake lava platform, comparing these with the KEGG database. The analysis identified 7 primary metabolic pathways and 40 secondary metabolic functions (Figure 3).

In primary metabolic pathways, the samples exhibited the following ranking of functional pathway abundances from highest to lowest: metabolism (50.83–51.44%), genetic information processing (18.61–19.48%), environmental information processing (12.93–13.14%), unclassified (11.96–12.44%), cellular processes (3.52–3.81%), human diseases (1.12–1.25%), and biological systems (0.82–0.90%). The soil bacterial population in the research area was dominated by metabolism. The metabolism and environmental information-processing pathways were much more prevalent in the GL than in the DB (*p* < 0.05), while the genetic information-processing and cellular processes pathways were significantly more prevalent in the DB than in the GL (*p* < 0.05).

We annotated 22 secondary metabolic functions with relative abundances exceeding 1%. A histogram was constructed for the top 10 functions. Notably, genes linked to amino acid metabolism (10.66–10.84%), membrane transport (10.17–10.88%), carbohydrate metabolism (10.03–10.40%), replication and repair (8.08–8.35%), and energy metabolism (6.00–6.15%) exhibited high relative abundances, representing key sub-functions of soil bacteria on the Jingpo lava platform. Within metabolic pathways, amino acids, carbohydrates, energy, cofactors and vitamins, and lipid metabolism mirrored the trends of membrane transport, an environmental information-processing function. Their relative abundances reached a zenith in the GL and exhibited considerable differences compared to those in the DB (*p* < 0.05). In the genetic information-processing pathway, the secondary functions of translation, replication, and repair exhibited significantly higher relative abundances in the DB compared to the other vegetation types (*p* < 0.05). This trend contrasted with the secondary functions in the metabolic and environmental information-processing pathways. Meanwhile, the secondary functions related to cell processes and signal transduction, categorized as unclassified, were not prominent, with their peak values observed in the CB.

The relative abundances of the metabolism and membrane transport pathways in the GL were notably high, aligning with the early succession law of ecosystems. Conversely, in the CF, the intricate litter delivery rate and decomposition process resulted in the predominance of genetic information-processing pathways.

### 3.5. Correlation Between Soil Physico-Chemical Properties and Bacterial Community Composition in Different Vegetation Types

The BD and SWC exhibited significant or highly significant positive correlations with Acidobacteriota and Planctomycetota (*p* < 0.05 or *p* < 0.01). The SOM, TN, TP, and TK were significantly or highly significantly positively correlated with Proteobacteria and Verrucomicrobiota (*p* < 0.05 or *p* < 0.01) and highly significantly negatively correlated with Actinobacteriota, Chloroflexi, Firmicutes, and Gemmatimonadota (*p* < 0.01). The pH was significantly or highly significantly positively correlated with Actinobacteriota, Chloroflexi, and Firmicutes (*p* < 0.05 or *p* < 0.01). AN and AK showed significant or highly significant positive correlations with Acidobacteriota, Chloroflexi, and Firmicutes (*p* < 0.05 or *p* < 0.01) and highly significant negative correlations with Proteobacteria and Verrucomicrobiota (*p* < 0.01) (Figure 4).

At the OTU level, an RDA analysis was performed on the soil physico-chemical properties and bacterial phylum level community structure for different vegetation types (Figure 5). RDA1 and RDA2 described 48.05% and 21.17% of the differences in the soil bacterial community structure among the five vegetation types, respectively. The driving factors that significantly affected the bacterial community structure were the SOM (R^2^ = 0.95, *p* = 0.001), BD (R^2^ = 0.90, *p* = 0.001), SWC (R^2^ = 0.89, *p* = 0.001), EC (R^2^ = 0.89, *p* = 0.001), and TN (R^2^ = 0.87, *p* = 0.001), and they are the key factors driving the structure of the soil bacterial community. The BD was the key driver of the soil bacterial community structure in the SL and the EC, SWC, TK, SOM, TP, TN, AN, and AK were the drivers in the DB (Figure 5).

### 3.6. Correlation Between Alpha-Diversity Indices of Soil Bacterial Communities and Soil Physico-Chemical Properties in Different Vegetation Types

A correlation analysis showed that the observed OTUs and the Chao1 index of the soil bacterial community were highly significant positively correlated (*p* < 0.01) with the SOM, TN, and AN, and significantly positively correlated (*p* < 0.05) with the BD, SWC, and AK. The Shannon index was highly significant positively correlated (p < 0.01) with the TN, TK, and AN and significantly positively correlated (*p* < 0.05) with the SOM and TP. The Pielou_e index was significantly positively correlated (*p* < 0.05) with the TP (Figure 6).

### 3.7. Correlation Between Soil Bacterial Community Functions and Soil Physico-Chemical Properties in Different Vegetation Types

As depicted in Figure 7, cell processes and signal transduction were highly positively correlated with the SWC, TN, SOM, and TK (*p* < 0.01), positively correlated with the EC and AN (*p* < 0.05), and negatively correlated with the BD and pH (*p* < 0.05). Translation was positively correlated with the EC, SWC, and TN (*p* < 0.05) and negatively correlated with the BD (*p* < 0.01). Lipid metabolism showed a strong positive correlation with the BD (*p* < 0.01) and a negative correlation with the SWC, EC, TN, and SOM (*p* < 0.05). Amino acid metabolism, carbohydrate metabolism, and membrane transport functions were highly similar in their effects on the soil properties, all being negatively correlated with the SWC, TN, SOM, and TK (*p* < 0.01). These findings indicate that the soil properties significantly influence bacterial community functions, with the BD, EC, SWC, TN, SOM, and TK as key factors. A cluster analysis revealed that amino acid metabolism, carbohydrate metabolism, and membrane transport were closely grouped, suggesting significant metabolic associations among them.

## 4. Discussion

### 4.1. Physico-Chemical Properties of Soils of Different Vegetation Types

Assessing the physical and chemical properties of soil is crucial for evaluating the soil’s quality and fertility [26]. Volcanic features, such as the eruption duration and the soil-forming matrices of volcanoes, have an impact on the plants found on lava terrace habitats. Together, these elements have a major effect on the soil’s physico-chemical characteristics [27]. This could be because soil nutrients are primarily determined by a variety of parameters, such as soil-forming matrices, the weathering severity, the leaching intensity, and topographical, climatic, and human effects [28]. The current study found that forest vegetation types in the Jingpo Lake lava plateau had a higher soil capacity than shrublands and grasslands. This was explained by the dispersion of plant roots, which were found primarily in the soil’s top layer in shrublands and grasslands. On the other hand, the plant roots of the different types of forest flora were well established and dispersed in a deeper layer, which was better for the soil’s compact structure. This phenomenon contradicts the dominant trend of changing the water content. It has been shown that the pore space and permeability of the soil are strengthened as a result of the bulk weight of the soil being reduced throughout the lava terrace vegetation restoration process [29]. Additionally, this improves the soil’s ability to retain water and decreases soil sloughing [30,31]. Biological influences, climate circumstances, human activities, and the parent material that forms the soil all have an impact on the pH of the soil. The reason for the lower pH values found in forest vegetation types in comparison to grasslands and shrublands is that forest vegetation has the ability to generate more apoplastic material. Coniferous species’ apoplastic substance has unique properties, such as being thicker than the cuticle and smaller than the leaf area. It is also distinguished by a greater concentration of waxes, tannins, resins, and other resistant materials. This substance decomposes more slowly than normal apoplastic material, and it becomes more acidic as it breaks down. The atmosphere turns acidic after the breakdown process [32,33]. It is clear that electrical conductivity is significantly impacted by the soil pH. Because of increased ionic dissociation, acidic soils typically have a better electrical conductivity. As a result, compared to other plant types, the electrical conductivity of forest vegetation types is significantly higher.

There was also a significant correlation between the SOM and the SWC, TN, TP, and TK because of a synergistic effect, or “coupling effect,” between changes in the SOM and changes in soil elements like N, P, and K. Additionally, the SOM, TN, TP, and TK of various vegetation types in the study area were in the order of coniferous forests > mixed coniferous and broad forests > deciduous forests > shrublands > grasslands. Growth is intimately linked to the elements N, P, and K in plants. Plant growth is strongly correlated with the elements N, P, and K. Additionally, the root systems of various vegetation types have varying capacities to absorb nutrients from the soil, and the types and quantities of apoplastic materials they produce differ [34], which results in variations in the nutrient content of the soil. It is clear that forest vegetation species have well-developed root systems that can generate large amounts of apoplastic materials with complex compositions, a strong restitution, and a high concentration of soil microorganisms. These microbes facilitate the transformation of materials and the circulation of energy, which is consistent with the findings of earlier research [29,32]. It has been shown that forest vegetation varieties have higher quantities of soil quick-acting nutrients. Plant development, metabolic interactions between the root system and soil microbes, and the breakdown of plant leftovers all contribute to this phenomenon. Together, these elements encourage the transformation of insoluble potassium into soluble potassium that acts quickly. As a result, the amount of soil quick-acting potassium rises as soluble quick-acting potassium rises [35]. The physical characteristics of the soil, the composition of the plant community, the microbial activity, and the effects of plants on one another’s roots all work together to control the amount of quick-acting phosphorus, which maps, to some extent, the reserve and supply status of soil phosphorus [36]. Soil phosphorus is found in the form of phosphate, which is readily absorbed by plants. According to certain studies, soil AP is higher on slopes with forest vegetation types than in grasslands [37]. These results might be explained by the different ways that various vegetation types affect the hydrological processes of slope erosion, which in turn causes changes in the distribution and content of soil AP. One of the factors contributing to the greater levels of AP in forest vegetation is the enhanced storage and supply capacity of AP that results from vegetation succession [38,39,40]. In conclusion, the enhancement of the soil physico-chemical characteristics during vegetation restoration in lava terrace ecosystems is a result of reciprocal limitations and cooperative action rather than autonomous activity.

### 4.2. Structure and Diversity of Soil Microbial Communities in Different Vegetation Types

Soil bacteria are vital to ecosystems, influencing plant community dynamics, soil organic matter processes, biogeochemical cycles, and ecosystem responses to environmental changes [41]. The structural attributes of these bacterial communities serve as key indicators of soil quality [10,42,43]. Our study reveals that, while dominant bacterial groups remain consistent across vegetation types, their relative abundances differ markedly, aligning with previous findings. The vegetation type has emerged as a crucial factor shaping soil microbial communities. In the study area, the phyla Actinobacteria, Proteobacteria, and Acidobacteria were notably abundant across all vegetation types, corroborating the results of Bai Xiaoxu, Liu Xing, and others [44,45]. Proteobacteria are adept at degrading complex lignin and cellulose, while Acidobacteria significantly contribute to the metabolism of plant residue carbon compounds and polymer degradation. The variation in the dominant phyla across vegetation types can be attributed to differing life-history strategies [43,46]. Proteobacteria, as copiotrophs, thrive in soils rich in organic matter and nutrients [47]. Conversely, Actinobacteria and Acidobacteria are oligotrophs, adapted to low-organic-matter environments [27]. In microbial ecology, copiotrophs align with the R-strategy, while oligotrophs align with the K-strategy [47,48]. This study’s findings underscore the pivotal role that these bacterial phyla play in predicting nutrient cycling and soil ecological quality in lava plateau ecosystems. Enhanced soil nutrients are correlated with an increased abundance of eutrophic bacteria [47]. Studies have demonstrated that the TN in soil is intricately associated with the composition and variety of bacterial communities. The presence of copiotrophic Proteobacteria in mixed coniferous and deciduous forests, as well as in coniferous forests, indicates elevated nutrient levels. The prevalence of Proteobacteria exhibits a positive link with the SOM [47,49,50] and a substantial negative correlation with the pH [51,52]. In contrast, oligotrophic Actinobacteria demonstrate a notable negative connection with both the soil organic matter and the total nitrogen, consistent with the results of this study [52,53,54].

Soil bacteria, the most dynamic entities within the soil ecosystem, play a crucial role in material cycling and energy flow, demonstrating a high sensitivity to environmental changes [55]. Variations in the composition and diversity of soil bacterial communities serve as key indicators of these changes, often quantified using microbial diversity indices [56]. This study revealed that the observed OTU, Chao1, and Shannon indices were significantly higher in coniferous forests compared to other vegetation types [57,58]. Changes in the soil fertility significantly impact the soil bacterial diversity. Different vegetation types foster varying nutritional strategies among bacterial groups, underpinning their niche differentiation. In coniferous forests, the complex canopy structure creates diverse microhabitats with varying light and heat conditions, accommodating bacteria with different light adaptations. In mixed coniferous–deciduous forests, the litter input results in carbon source gradient differentiation. Litter from broad-leaved trees with a low C/N ratio accelerates organic matter decomposition, whereas coniferous litter, rich in lignin and cellulose, contributes to a stable organic carbon pool. This stratification into litter, humus, and mineral soil layers supports distinct bacterial functional groups, minimizing niche overlap [59]. Wu Jinzhuo et al. reported that the soil bacterial diversity in mixed coniferous–deciduous forests is significantly higher than in monocultures [54,60]. This suggests that the chemical uniformity of litter and root exudates in single-species stands reduces the substrate diversity, limiting bacterial niche differentiation. Additionally, the shallow roots of herbaceous plants restrict the nutrient availability for deep-soil microorganisms, compromising the soil organic carbon stability. The depth of vegetation root distribution appears to be a crucial biological factor affecting the bacterial community diversity [61,62]. This study found no significant difference in the community evenness, as indicated by the Pielou’s index, suggesting that soil bacterial communities on the Jingpo Lake lava platform employ a balanced resource allocation strategy across various vegetation types [12]. The unique physical and chemical properties of soil derived from volcanic rock, including a high porosity that enhances gas diffusion and iron and magnesium ions from basalt weathering, help maintain community stability by regulating extracellular enzyme activity.

### 4.3. Potential Functional Groups of Soil Microorganisms

Soil bacteria exhibit similar metabolic pathways across various vegetation types, with the vegetation influencing the bacterial community structure through both its above- and below-ground components, thereby impacting bacterial functions. This study demonstrated that soil bacteria primarily employ metabolic pathways to sustain ecosystem stability, aligning with the findings of Tong Xiaodong, Liu Zhu, and others [63,64,65]. Secondary functions involve deamination and transamination during amino acid metabolism, converting amino acids into amines, keto-acids, and CO_2_, processes integral to the nitrogen cycle in soil and plants [66]. Carbohydrate metabolism, through the tricarboxylic acid cycle, glycolysis, and pentose phosphate pathway, generates energy and metabolites that support plant and bacterial growth and development [67,68]. Carbohydrate metabolism is integral to the cycling of essential nutrients like nitrogen, phosphorus, and potassium in plants, thereby enhancing nutrient cycling. Soil bacteria contribute to the energy flow and material cycling through metabolic processes, improving the soil conditions and regulating plant growth. This explains the high relative abundance of metabolic functions such as amino acid, carbohydrate, and energy metabolism [64]. However, secondary functional pathways, including membrane transport, replication and repair, and translation, also showed a high abundance, contrasting with the findings by Jiao Yue et al. [69]. This discrepancy may stem from the spatiotemporal variations due to differing geological conditions across study areas. The Jingpo Lake lava platform is undergoing vegetation recovery; it exhibits weaker metabolic functions compared to other geological settings, yet possesses enhanced repair and treatment capabilities. Additionally, the lava platform’s soil demonstrates a high level of microbial activity, close interpopulation relationships, and a stable community structure, leading to increased membrane transport and translation [70].

Vegetation restoration on the lava plateau markedly affects soil bacterial functions. Metabolic pathways and secondary functions are most pronounced for meadow vegetation, likely due to the early-stage abundance of soil nutrients. This abundance enables microbial communities to swiftly decompose simple organic matter, absorb nutrients, and support pioneer plant growth [71]. As restoration advances to forest vegetation, soil microorganisms prioritize DNA maintenance and synthesis, adopting a K-strategy for litter input to decompose complex, recalcitrant organic matter. Consequently, genetic information-processing pathways and their secondary functions are more prevalent than in other vegetation types. In forest ecosystems, as succession progresses, the functional abundance of the soil bacteria generally rises, indicating an increased physiological activity, ongoing material and energy cycling, and movement towards functional optimization [72].

Energy metabolism and membrane transport functions were most prevalent in the GL, but significantly decreased in the DB (*p* < 0.05) [35,41,73]. In early succession stages, the simple microbial community structure in grasslands leads to higher abundances of metabolic and environmental information-processing pathways, with a specific trend in vegetation. As succession progresses to forest vegetation, genetic information-processing pathways increase, indicating an enhanced microbial adaptability to complex substrates and stable habitats. In the coniferous–broad-leaved mixed forest, cell processes and signal transduction peaked, reflecting a greater microbial functional diversity. This suggests that forest ecosystems rely more on precise genetic information processing and stable energy supply mechanisms [72,74].

Different vegetation types affect soil bacterial functions by altering the soil nutrients and the microecological environment. Grasslands are characterized by fast microbial metabolism (R-strategy), while forest ecosystems rely on sophisticated genetic processing and stable energy sources (K-strategy). Moreover, as succession progresses, bacterial functions typically grow more complete and complex.

## 5. Conclusions

The co-evolutionary dynamics of the vegetation–soil–microbiome system were investigated in the unique geological context of the Cenozoic volcanic lava plateau surrounding Jingpo Lake. Vegetation at different successional stages modulated key edaphic drivers such as the soil organic matter, available nitrogen, total nitrogen, and soil water content through root exudation and litter decomposition, thereby shaping distinct microbial community structures and functional attributes. Actinobacteriota, Proteobacteria, and Acidobacteria established a material cycling system centered around amino acid and carbohydrate metabolism in the lava platform soils, creating a microenvironment conducive to the proliferation of dominant genera like Candidatus_Udaeobacter. The intricate interactions among vegetation, soil, and microorganisms accelerated the primary pedogenesis of the volcanic lava, with nutrient retention and metabolite release promoting the ecological succession of plant communities. Lava plateau ecosystems undergo a vegetation reconstruction process, during which the rhizosphere soil harbors abundant bacterial resources. Forest vegetation types acidify the soil through more complete litter decomposition, promoting the colonization of copiotrophic bacterial communities dominated by Proteobacteria. This balances the functional diversity and nutrient retention, achieving a stable positive-feedback restoration cycle. Strengthening the application of forest vegetation is crucial for the adaptive management of lava plateaus. Plants do not passively adapt to the lava plateau soil environment; rather, they gradually adapt to extreme habitats and reconstruct ecosystem functions through the plant–soil–microorganism coupling mechanism. This provides a new scientific perspective for exploring the restoration mechanism of lava plateau ecosystems and offers theoretical references for vegetation reconstruction and the mechanism of plant niche construction in volcanic landform areas.

Future research should focus on validating the current functional predictions from PICRUSt2 using enzyme activity and metagenomic sequencing. Additionally, integrating plant–microbe interaction mechanisms with a fungal community analysis will enhance our understanding of ecosystem restoration. Cross-seasonal dynamic monitoring and comparative studies across various volcanic landforms will offer broader applicability. These efforts are expected to provide a scientific foundation for vegetation restoration mechanisms and the adaptive management of unique ecosystems globally.

## Figures and Tables

**Figure 1 microorganisms-13-01648-f001:**
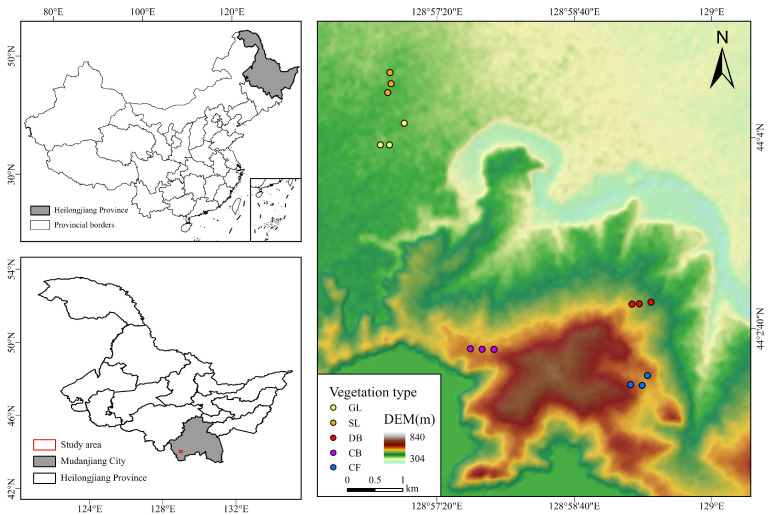
Study area of Jingpo Lake World Geopark, Heilongjiang Province, China.

**Figure 2 microorganisms-13-01648-f002:**
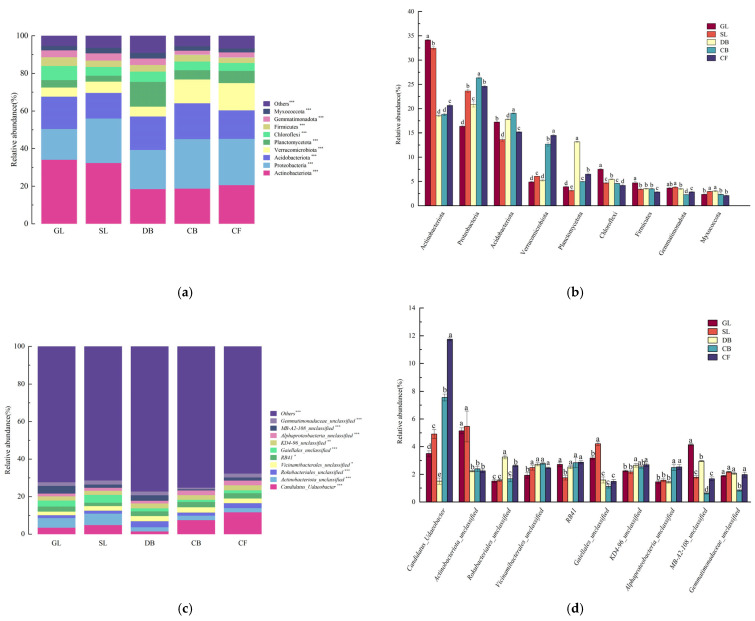
Different letters indicate significant differences at the 0.05 level; Soil bacterial community composition in different vegetation types: (**a**) community composition of bacterial phyla in different vegetation types; (**b**) horizontal distribution of bacterial phyla in different vegetation types; (**c**) community composition of bacterial genera in different vegetation types; and (**d**) horizontal distribution of bacterial genera in different vegetation types. (*** *p* < 0.001), (** *p* < 0.01), and (* *p* < 0.05) indicate significant differences in abundance among soil microorganisms in the five vegetation types according to the Wilcoxon signed-rank test.

**Figure 3 microorganisms-13-01648-f003:**
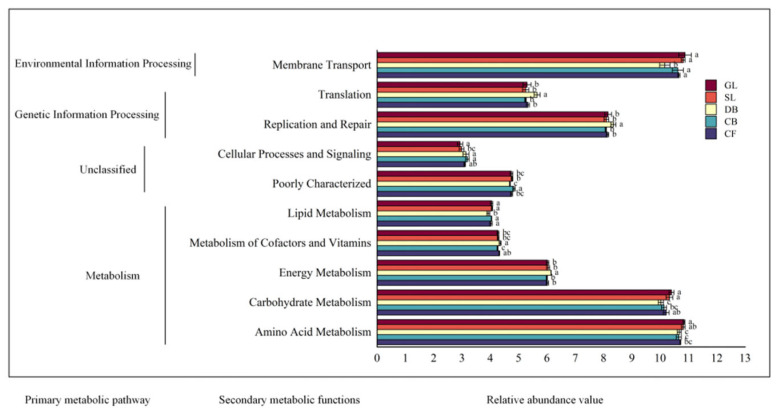
Different letters indicate significant differences at the 0.05 level; Predictions of soil bacterial community function for different vegetation types based on the KEGG database (reflecting the top 10 species with a relative abundance greater than 1%).

**Figure 4 microorganisms-13-01648-f004:**
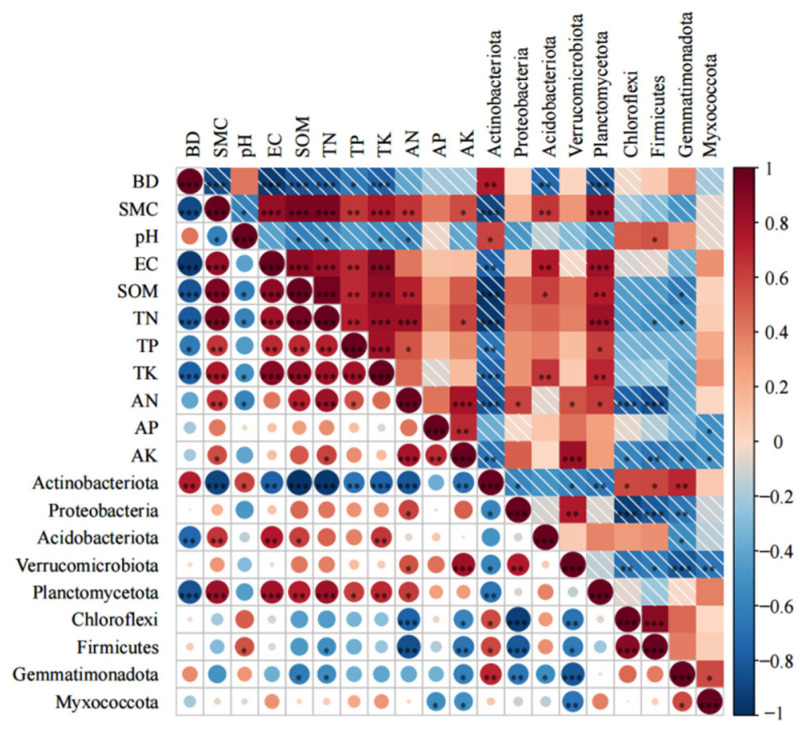
Integrated analysis of soil–microbe correlations using a heatmap to show FDR-corrected correlations between bacterial phyla (rows) and soil properties (columns). Colored circles indicate Pearson’s r values (blue: negative, red: positive; scale shown). Symbols denote significance: * *p* < 0.05, ***p* < 0.01, ****p* < 0.001.

**Figure 5 microorganisms-13-01648-f005:**
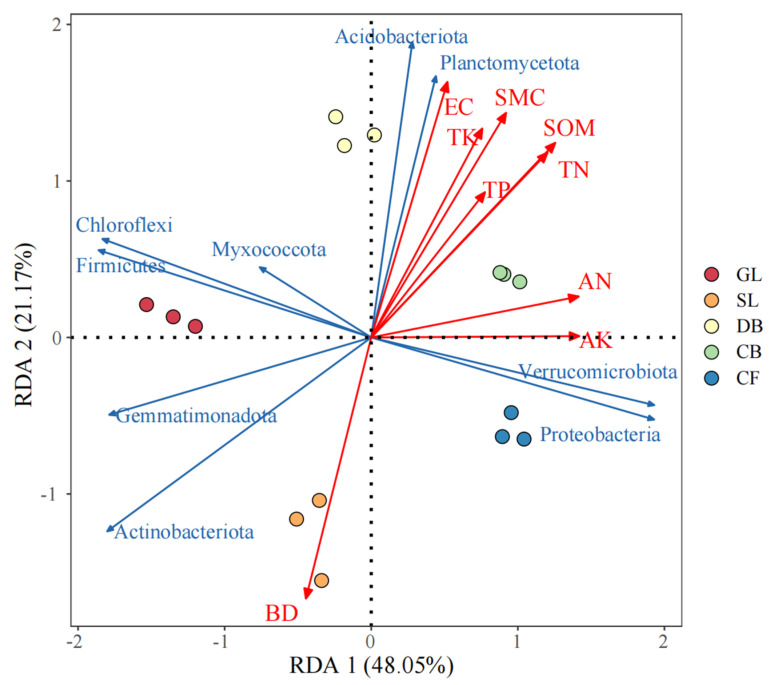
RDA analysis of soil physico-chemical properties and bacterial community composition based on phylum level. Ordination diagram showing the effect of environmental variables (red line, red arrows) on the sample locations (circles). RDA triplot with environmental factors (red arrows), microbial communities (blue dots), and vegetation types (green labels). Statistical significance of RDA axes: Axis 1 (*p* = 0.003) and Axis 2 (*p* = 0.021) after FDR correction. Abbreviations: BD, bulk density; SWC, soil water content; SOM, soil organic matter.

**Figure 6 microorganisms-13-01648-f006:**
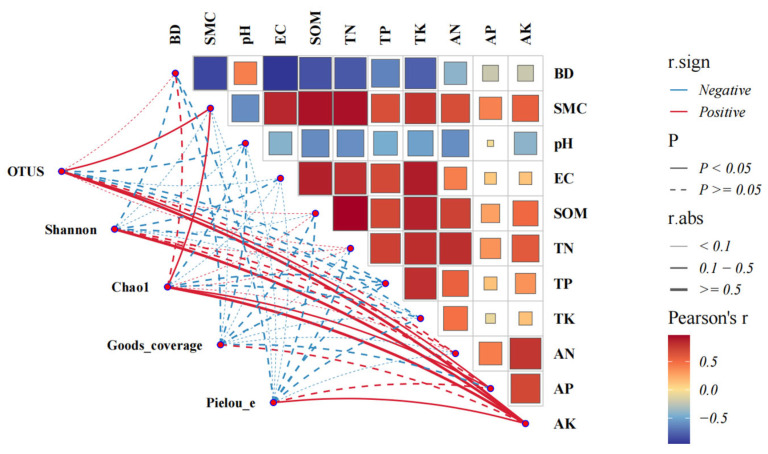
Correlation between soil bacterial community diversity and physico-chemical properties in different vegetation types.

**Figure 7 microorganisms-13-01648-f007:**
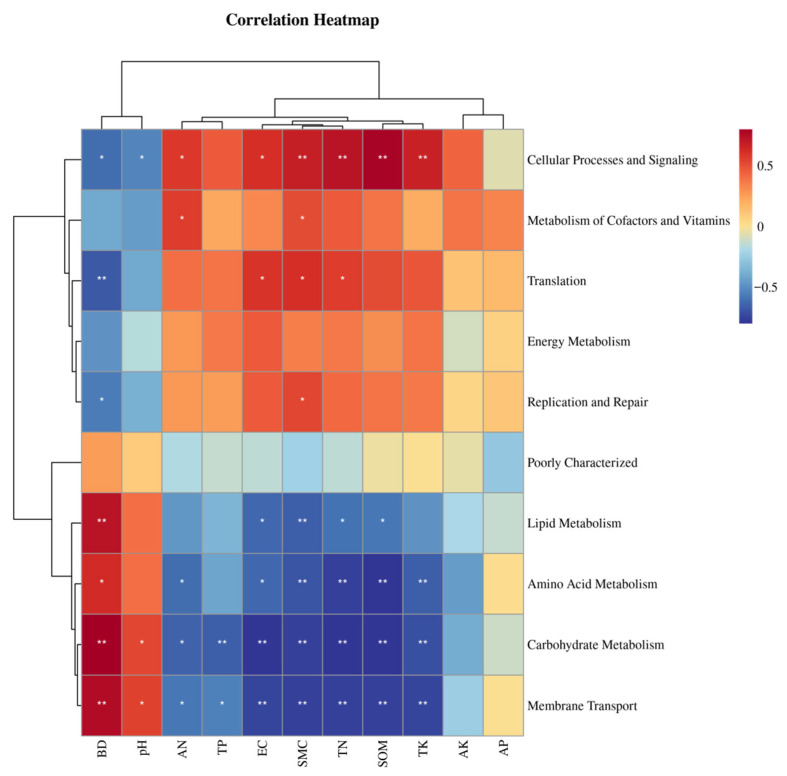
Correlation between soil bacterial community functions and soil physico-chemical properties in different vegetation types. Symbols denote significance: * *p* < 0.05, ***p* < 0.01.

**Table 1 microorganisms-13-01648-t001:** Dominant plant species (based on field survey criteria).

Vegetation Types	Longitude and Latitude	Dominant Species
GL	128°94′79″ E–44°06′58″ N	*Artemisia annua*, *Carex duriuscula*, *Potentilla chinensis*, *Galium verum*
	128°95′03″ E–44°06′83″ N	
	128°94′64″ E–44°06′58″ N	
SL	128°94′76″ E–44°07′18″ N	*Lespedeza davurica*, *Lonicera japonica*, *Rhamnus davurica*, *Acer tataricum* subsp. *ginnala*
	128°94′80″ E–44°07′41″ N	
	128°94′81″ E–44°07′29″ N	
DB	128°96′10″ E–44°04′21″ N	*Quercus mongolica*, *Populus davidiana*, *Albizia kalkora*, *Prunus mandshurica*, *Betula platyphylla*
	128°96′47″ E–44°04′20″ N	
	128°96′28″ E–44°04′20″ N	
CB	128°99′02″ E–44°04′75″ N	*Pinus koraiensis*, *Abies nephrolepis*, *Tilia amurensis*, *Quercus mongolica*
	128°98′71″ E–44°04′725″ N	
	128°98′83″ E–44°04′73″ N	
CF	128°98′68″ E–44°03′79″ N	*Larix gmelinii*, *Abies nephrolepis*, *Picea koraiensis*, *Picea jezoensis*
	128°98′96″ E–44°03′90″ N	
	128°98′87″ E–44°03′78″ N	

**Table 2 microorganisms-13-01648-t002:** Soil physico-chemical characteristics of different vegetation types in the lava plateau.

Sample	GL	SL	DB	CB	CF
BD (g·cm^−3^)	1.92 ± 0.06 ^b^	2.33 ± 0.24 ^a^	1.88 ± 0.11 ^b^	1.49 ± 0.18 ^c^	1.01 ± 0.11 ^d^
SWC (%)	35.57 ± 1.40 ^c^	23.22 ± 2.50 ^d^	52.12 ± 4.26 ^b^	53.97 ± 3.45 ^b^	66.75 ± 3.56 ^a^
pH	6.74 ± 0.21 ^a^	6.50 ± 0.22 ^ab^	6.34 ± 0.35 ^ab^	6.31 ± 0.19 ^ab^	6.23 ± 0.30 ^b^
EC (μs·cm^−1^)	101.21 ± 4.20 ^c^	95.57 ± 2.08 ^d^	102.58 ± 1.76 ^c^	122.30 ± 3.81 ^b^	133.84 ± 1.40 ^a^
SOM (g·kg^−1^)	185.41 ± 2.22 ^d^	188.17 ± 2.57 ^d^	225.55 ± 2.57 ^c^	246.20 ± 2.52 ^b^	253.01 ± 2.21 ^a^
TN (g·kg^−1^)	5.73 ± 0.16 ^c^	5.88 ± 0.06 ^c^	7.58 ± 0.21 ^b^	7.66 ± 0.15 ^b^	8.49 ± 0.39 ^a^
TP (g·kg^−1^)	0.86 ± 0.11 ^b^	0.96 ± 0.14 ^ab^	1.04 ± 0.05 ^ab^	1.13 ± 0.22 ^ab^	1.25 ± 0.19 ^a^
TK (g·kg^−1^)	10.37 ± 0.15 ^b^	10.57 ± 0.06 ^b^	10.63 ± 0.21 ^b^	11.53 ± 0.31 ^a^	11.77 ± 0.32 ^a^
AN (mg·kg^−1^)	219.39 ± 4.69 ^e^	330.75 ± 4.83 ^d^	474.72 ± 5.98 ^a^	362.28 ± 5.28 ^c^	440.91 ± 7.15 ^b^
AP (mg·kg^−1^)	20.18 ± 1.44 ^ab^	17.25 ± 1.74 ^b^	23.39 ± 2.29 ^a^	19.37 ± 1.89 ^b^	20.28 ± 1.09 ^ab^
AK (mg·kg^−1^)	126.36 ± 4.12 ^c^	125.42 ± 3.25 ^c^	175.38 ± 6.29 ^a^	145.51 ± 7.08 ^b^	142.18 ± 1.95 ^b^

Note: Different letters in the same column represent significant differences at the 0.05 level; the same is true below. BD (soil bulk density), SWC (soil water content), EC (electrical conductivity) SOM (soil organic matter), TN (total nitrogen), TK (total potassium), TP (total phosphorus), AN (alkaline dissolved nitrogen), AP (immediate phosphorus), AK (immediate potassium).

**Table 3 microorganisms-13-01648-t003:** Characteristics of soil bacterial richness (observed OTUs) and diversity index under different vegetation types.

	Observed OTUs	Shannon	Chao1	Pielou-e	Coverage
GL	2568.67 ± 54.90 ^b^	10.36 ± 0.17 ^b^	2612.18 ± 71.93 ^bc^	0.92 ± 0.01 ^a^	0.97 ± 0.01 ^a^
SL	2269.33 ± 7.57 ^c^	10.28 ± 0.04 ^b^	2274.20 ± 8.73 ^d^	0.92 ± 0.01 ^a^	0.98 ± 0.01 ^a^
CF	3013.00 ± 128.27 ^a^	10.84 ± 0.24 ^a^	3001.61 ± 81.28 ^a^	0.93 ± 0.01 ^a^	0.97 ± 0.03 ^a^
CB	2628.33 ± 60.58 ^b^	10.47 ± 0.09 ^b^	2712.88 ± 49.35 ^b^	0.92 ± 0.01 ^a^	0.97 ± 0.01 ^a^
DB	2548.67 ± 2.08 ^b^	10.32 ± 0.23 ^b^	2517.81 ± 57.91 ^c^	0.92 ± 0.01 ^a^	0.98 ± 0.02 ^a^

Note: Different letters in the same column represent significant differences at the 0.05 level.

## Data Availability

The original contributions presented in this study are included in the article. Further inquiries can be directed to the corresponding author.

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
