# Peer review of "Characterization of Soil Bacterial Communities in Different Vegetation Types on the Lava Plateau of Jingpo Lake"

_microorganisms, 2025, doi:10.3390/microorganisms13071648_

Round 1
Reviewer 1 Report
Comments and Suggestions for Authors
Characterisation of soil bacterial communities in different
vegetation types on the lava Plateau of Jingpo Lake
- General comments
The manuscript has great potential, it could be interesting for the readers, fits to the scope of Renewable Energy. However, the AI index is 30% and these numbers may be problematic regarding publishing, they should be reduced, the AI should also be explained, if remain some part. Literature sources are relevant, methodology is high level, results are remarkable, well-supported and can contribute to the development of the scope, however, it contains some minor inaccuracies, which needs clarification, or explanation. I hope that my comments may help for the authors to make the manuscript better and suitable for publication. The professional content is high level, the reason of the major revision is the high similarity and artificial intelligency index and the missing citation detailed in specific comments.
- Lack of clarity on experimental design. Although five types of vegetation are analysed, details on randomisation, sampling depth, management of environmental bias (slope, shade, etc.) are missing. You should add a paragraph on the experimental protocol: 1. sampling depth and name of regions and geographical coordinates; 2. justification for the choice of plots. 3. method to avoid pseudo-replication.
- Some of the keywords (ex: lava plateau) should not be the same as the words in the title. Differential expressions may result more shots via search engines and make the future citations easier.
- Some minor formal inaccuracies could be improved
- Text in figures 1 is unreadable.
- Lack of a map of the sites with comprehensible legends.Figure 1 is mentioned but is not clearly visible and the text in china. Add a high-resolution map with a clear legend indicating vegetation types and altitude.
- Specific comments
Section 3.2. This section is intended to present results on bacterial communities based on 16S rRNA gene sequencing, which is a molecular marker specific to prokaryotes (bacteria). However, the authors repeatedly mention fungal taxa in this section, including: Ascomycota, Aspergillus, Mucor. This is a significant taxonomic misclassification. These taxa are fungi, not bacteria, and they cannot be accurately detected using 16S rRNA sequencing, which does not target the eukaryotic rRNA genes such as 18S or ITS regions.
1. Issue in Section 3.2: “Soil Bacterial Community Composition in Different Vegetation Types”. Check the taxonomic database used (SILVA, Greengenes?) and correct any erroneous names. These errors call into question the validity of the taxonomic analysis.
2. There is no mention in the manuscript of the raw sequencing data being deposited in a public repository such as NCBI SRA. For transparency and reproducibility, it is essential to deposit all sequencing data in a publicly accessible database and provide the accession number(s) in the manuscript.
3. “The subsequent PCR products were then amplified using AMPureXTbeadss": This is incorrect. AMPure XP beads are used for purification, not amplification!!!!!!!!!.
4. The following information should be added: Read length (e.g., 2×250 bp?) + Sequencing depth (how many reads per sample on average?) + Number of samples sequenced + Were negative controls included?
- The paragraph ends with sequencing details but omits any information on the downstream processing of the sequence data, which is critical to validate the results.
Suggestion: Add a detailed description of the bioinformatics pipeline used, including:
- Which software was used (e.g., QIIME2, DADA2)?
- How were sequences denoised or clustered (ASV vs OTU)?
- Which reference database was used for taxonomic assignment (SILVA, Greengenes)?
- What quality filtering thresholds were applied
6. The phrase “OTUS species number” is not scientifically accurate. OTUs (Operational Taxonomic Units) are not the same as species. Use “Number of OTUs” or “Observed OTUs”
7. Weak biological/ecological interpretation of results. The section lists percentages and statistical differences between vegetation types, but does not interpret what these functions mean in an ecological context.
8. Poor structure and readability. Long and repetitive sentences make it hard to follow the logic of comparisons between vegetation types. Break this section into clear sub-sections:
- Overview of predicted functions
- First-level pathway differences
- Second-level pathway differences
- Notable vegetation-specific trends
9. Multiple subsections repeat that SOM, TN, TP, TK, etc. are correlated with many bacterial phyla or diversity indices, often with similar phrasing. Consolidate these results with an integrated correlation matrix or summary table and Focus on highlighting key or unexpected findings rather than listing all.
- Section 3.7 lists many correlations between soil properties and bacterial functions without clear organization. Break the section into subsections by function group (e.g., metabolic pathways, information processing, etc.), Present data in visual form (e.g., clustered heatmap, function-environment biplots) and Focus on key functional insights — for example, why does higher SOM reduce carbohydrate metabolism? Is this expected?
- The discussion is often limited to repeating the results, without any advanced ecological interpretation. Refocus the discussion around two or three main hypotheses, for example: bacterial richness increases with the structural complexity of the vegetation; or certain bacterial communities are indicators of primary succession.
- Lack of originality in the conclusion. The conclusion is a repetition of the results and its preferable to add a concrete perspectives: how can these results be used for ecological restoration? And what are the useful indicator taxa for monitoring succession?
- Some references are from Master's theses, which are not peer-reviewed and thus not recommended for scientific publications.

Author Response
Dear Reviewer:
Thank you very much for taking the time to review this manuscript. We have revised the manuscript according to your suggestions. Please find the detailed responses below.
Point-by-point response to Comments and Suggestions for Authors |
Comments 1:Lack of clarity on experimental design. Although five types of vegetation are analysed, details on randomisation, sampling depth, management of environmental bias (slope, shade, etc.) are missing. You should add a paragraph on the experimental protocol: 1. sampling depth and name of regions and geographical coordinates; 2. justification for the choice of plots. 3. method to avoid pseudo-replication. Response 1:We sincerely appreciate your valuable suggestions. We have supplemented the text with information on the reasons for selecting the sample sites, the depth of soil sample collection, and specific geographical coordinates. The specific location is on page 3, Chapter 2, Materials and Methods, Section 2, Soil Sample Collection, lines 108 to 117. |
|
Comments 2:Some of the keywords (ex: lava plateau) should not be the same as the words in the title. Differential expressions may result more shots via search engines and make the future citations easier. Response 2:We sincerely appreciate your valuable suggestions, and we have refined the keywords. The specific location is on page 1, keywords section, lines 30 to 31. |
|
Comments 3:Some minor formal inaccuracies could be improved. Response 3:We thank the experts for their valuable comments, and we will revise it carefully to further improve the quality of the paper. For some expressions in the text that are not rigorous or accurate enough, we will improve them one by one in the revision. |
|
Comments 4:Lack of a map of the sites with comprehensible legends.Figure 1 is mentioned but is not clearly visible and the text in china. Add a high-resolution map with a clear legend indicating vegetation types and altitude. Response 4:We thank the reviewers for their valuable comments. We have noted that Figure 1 is deficient in terms of clarity and legend description. In the revised draft, we will add a high-resolution map with a clear legend and clearly labeled vegetation types and elevations to enhance the readability and scientific quality of the diagram.Furthermore, the specific latitude and longitude of the sampling points are supplemented in Table 1. The specific locations are shown in Figure 1 Study area of Jingpo Lake World Geopark, Heilongjiang Province, China on page 3, Chapter 2, Materials and Methods, Section 1 Site Description and Soil Sampling, lines 105 to 106; and Table 1 Dominant Plant Species (Based on Field Survey Criteria) on page 4, Chapter 2, Materials and Methods, Section 2 Soil Collection and Treatment, line 118. Thank you again for your suggestions! |
|
Comments 5:Issue in Section 3.2: “Soil Bacterial Community Composition in Different Vegetation Types”. Check the taxonomic database used (SILVA, Greengenes?) and correct any erroneous names. These errors call into question the validity of the taxonomic analysis. Response 5:We thank the reviewers for their valuable comments. We apologize for our carelessness and have corrected the fungal phylum to bacterial phylum in our resubmitted manuscript. Thank you for your correction. The specific location is on page 6, Chapter 3, Results and Analysis, Section 2 Soil Bacterial Community Composition in Different Vegetation Types, lines 196 to 206. |
|
Comments 6:There is no mention in the manuscript of the raw sequencing data being deposited in a public repository such as NCBI SRA. For transparency and reproducibility, it is essential to deposit all sequencing data in a publicly accessible database and provide the accession number(s) in the manuscript. Response 6:We thank the reviewers for their valuable comments.We have deposited the raw sequencing data in the NCBI SRA public database and have supplemented the access number in the manuscript. The specific location is on page 5, Chapter 2, Materials and Methods, Section 4 High-Throughput Sequencing of Soil Bacterialines 152 to 153. |
|
Comments 7:“The subsequent PCR products were then amplified using AMPureXTbeadss": This is incorrect. AMPure XP beads are used for purification, not amplification!!!!!!!!!. Response 7:Thank you for your valuable comments, we apologize for our carelessness. In our resubmitted manuscript, the technical description error has been corrected to clarify that the AMPure XP magnetic beads are used for purification of PCR products rather than amplification, and the revised formulation is more in line with the actual process of the experimental operation to ensure the accuracy of the technical details. Thank you for your correction. The specific location is on page 4, Chapter 2, Materials and Methods, Section 4 Throughput Sequencing of Soil Bacterialines, paragraph 1, lines 137 to 138. |
|
Comments 8:The following information should be added: Read length (e.g., 2×250 bp?) + Sequencing depth (how many reads per sample on average?) + Number of samples sequenced + Were negative controls included? Response 8:Thank you for your valuable suggestions.The sequencing technical parameters (reading length, sequencing depth, sample number and negative quality control) were supplemented to ensure the repeatability of the experiment. The specific location is on page 4, Chapter 2, Materials and Methods, Section 4 Throughput Sequencing of Soil Bacterialines , paragraph 1, lines 140 to 144. |
|
Comments 9:The paragraph ends with sequencing details but omits any information on the downstream processing of the sequence data, which is critical to validate the results. Suggestion: Add a detailed description of the bioinformatics pipeline used, including: Which software was used (e.g., QIIME2, DADA2)? How were sequences denoised or clustered (ASV vs OTU)? Which reference database was used for taxonomic assignment (SILVA, Greengenes)? What quality filtering thresholds were applied. Response 9:The process of bioinformatics analysis (software, denoising method, reference database, quality filtering threshold) is described in detail to improve the transparency of the method. The insertion position is selected at the end of the sequencing method section, which is logically coherent and convenient for readers to check the technical details. The specific location is on page 4, Chapter 2, Materials and Methods, Section 4 Throughput Sequencing of Soil Bacterialines, Second to fourth paragraphs, lines 145 to 162. |
|
Comments 10:The phrase “OTUS species number” is not scientifically accurate. OTUs (Operational Taxonomic Units) are not the same as species. Use “Number of OTUs” or “Observed OTUs”. Response 10:Thank you for your valuable feedback, we have changed the OTU to Observed OTU. |
|
Comments 11:Weak biological/ecological interpretation of results. The section lists percentages and statistical differences between vegetation types, but does not interpret what these functions mean in an ecological context. Response 11:Thank you for your valuable comments on this paper. Regarding the ecological interpretation problems you pointed out, we have made corresponding additions in the revised manuscript to enhance the discussion of the ecological significance of the results. Your suggestions are of great value to the improvement of our study. All changes are reflected in the discussion section of Chapter 4. |
|
Comments 12:Poor structure and readability. Long and repetitive sentences make it hard to follow the logic of comparisons between vegetation types. Break this section into clear sub-sections:Overview of predicted functions.First-level pathway differences.Second-level pathway differences.Notable vegetation-specific trends. Response 12:Thank you for your valuable suggestions.We have divided this part into clear sub-parts:Overview of prediction function.Primary pathway difference.Secondary pathway difference.Significant vegetation-specific trends. The specific location is on page 8, Section 4 of Chapter 3, ‘Prediction of Soil Bacterial Community Function in Different Vegetation Types,’ paragraphs 1 to 4, lines 239 to 271. |
|
Comments 13:Multiple subsections repeat that SOM, TN, TP, TK, etc. are correlated with many bacterial phyla or diversity indices, often with similar phrasing. Consolidate these results with an integrated correlation matrix or summary table and Focus on highlighting key or unexpected findings rather than listing all. Response 13:Thank you for your valuable suggestions. We have systematically integrated the relevant analysis results in the paper, removed redundant expressions, and highlighted key findings and unexpected association patterns through the integrative analysis tables. Your comments have made the paper more concise and focused. Thank you again for your careful review and constructive comments on this paper! Changes have been made throughout Chapter 3 Results and Analysis. |
|
Comments 14:Section 3.7 lists many correlations between soil properties and bacterial functions without clear organization. Break the section into subsections by function group (e.g., metabolic pathways, information processing, etc.), Present data in visual form (e.g., clustered heatmap, function-environment biplots) and Focus on key functional insights — for example, why does higher SOM reduce carbohydrate metabolism? Is this expected? Response 14:We sincerely appreciate your feedback. We have further broken down the results and analysis in Section 7 of Chapter 3, ‘Correlation between Soil Bacterial Community Functions and Soil Physico-Chemical Properties in Different Vegetation Types,’ into functional groups, and the specific insights into key functions are supplemented in Section 3 of Chapter 4, ‘Potential Functional Groups of Soil Microorganisms.’ The specific location is on page 11, Section 7 of Chapter 3, ‘Results and Analysis,’ titled ‘Correlation between Soil Bacterial Community Functions and Soil Physico-Chemical Properties in Different Vegetation Types,’ in the first paragraph, lines 322 to 334. Additionally, in Chapter 4, Section 3, ‘Potential Functional Groups of Soil Microorganisms,’ paragraphs 2 to 4, lines 482 to 508. |
|
Comments 15:The discussion is often limited to repeating the results, without any advanced ecological interpretation. Refocus the discussion around two or three main hypotheses, for example: bacterial richness increases with the structural complexity of the vegetation; or certain bacterial communities are indicators of primary succession. Response 15:We thank the reviewers for their valuable comments. We have revised the discussion section to avoid simple repetition of the results and strengthened the in-depth analysis and interpretation at the ecological level to better reflect the theoretical significance and ecological value of the findings. Thank you again for your suggestions! |
|
Comments 16:Lack of originality in the conclusion. The conclusion is a repetition of the results and its preferable to add a concrete perspectives: how can these results be used for ecological restoration? And what are the useful indicator taxa for monitoring succession? Response 16:We thank the reviewers for their pertinent comments. We have revised the conclusion section to further refine the innovations of the study, emphasize the ecological significance and theoretical value of the findings, and avoid simple repetition of the findings to enhance the originality and depth of the conclusion. The specific location is on page 8, Chapter 5 Conclusions, paragraphs 1 to 2, lines 522 to 542.Thank you again for your suggestions! |
|
Comments 17:Some references are from Master's theses, which are not peer-reviewed and thus not recommended for scientific publications. Response 17:Thank you for your valuable feedback regarding the references in our manuscript. We acknowledge that some citations are drawn from Master’s theses, which are not peer-reviewed, and we will replace them with more authoritative sources where possible. Your comment has helped us improve the rigor of our work, and we appreciate your thorough review. |

Reviewer 2 Report
Comments and Suggestions for Authors
1. Please revise the abstract to reduce background details and include more emphasis on the key findings and the significance of your study.
2. The first paragraph of the introduction needs clarification. Please revise it to focus only on content essential for framing your research. Additionally, clearly explain why you selected the five site types and elaborate on the research gap your study addresses.
3. Please discuss the limited existing research on the relationship between microbial communities and lava terraces.
4. Clarify how dominant species were defined in Table 1.
5. In section 2.3, please specify the chemical parameters analyzed and include the names and details of the measurements.
6. Indicate which DNA extraction kit was used and provide its full name and manufacturer.
7. Describe how you processed and normalized the 16S rRNA data, including the specific methods or software used.
8. Clarify whether your data met the assumptions required for ANOVA and chi-square tests.
9. Include a description of PICRUSt2 methodology in the Methods section.
10. Adjust the p-values shown in Figure 4, and consider using the FDR or Holm correction method for multiple comparisons.
11. Please add and adjust p-values for your RDA analysis to ensure statistical validity.
12. I recommend removing Figure 6, as it appears unnecessary. Consider integrating the diversity data into Figure 4 and presenting the results there.
13. For Figure 7, adjust the p-values accordingly. You may also consider removing this figure and incorporating relevant parameters into Figure 4 to avoid redundancy. The current manuscript includes too many figures that are not well justified. Moreover, the discussion lacks depth regarding microbial functional potential and diversity.
14. Expand the conclusion to highlight broader environmental implications, the key questions addressed, the importance of your findings, and limitations (e.g., inferring function based on 16S data).
15. The manuscript requires thorough language editing to improve clarity and readability prior to publication.
Author Response
Dear Reviewer:
Thank you very much for taking the time to review this manuscript. We have revised the manuscript according to your suggestions. Please find the detailed responses below.
Point-by-point response to Comments and Suggestions for Authors |
Comments 1:Please revise the abstract to reduce background details and include more emphasis on the key findings and the significance of your study. Response 1:We sincerely appreciate the reviewer's suggestion. I have streamlined the background content and highlighted the key findings and research significance. The exact location is on the first page, the first paragraph, lines 12 to 24. |
|
Comments 2:The first paragraph of the introduction needs clarification. Please revise it to focus only on content essential for framing your research. Additionally, clearly explain why you selected the five site types and elaborate on the research gap your study addresses. Response 2:We sincerely appreciate your suggestions and have made the following revisions: In the introduction, we revised the background information, specifically lines 34 to 40 of the first paragraph on the first page. We explained the reasons for selecting five types of vegetation, specifically lines 40 to 45 of the first paragraph on the first page. We explained the issues addressed in the study, specifically lines 88 to 92 of the third paragraph on the second page. |
|
Comments 3:Please discuss the limited existing research on the relationship between microbial communities and lava terraces. Response 3:We sincerely appreciate the reviewer's suggestion. We have added relevant research on soil microorganisms in lava plateaus. The specific location is on page 2, in the third paragraph of the introduction, lines 77 to 84. |
|
Comments 4:Clarify how dominant species were defined in Table 1. Response 4:We sincerely appreciate the reviewer's suggestion. In the Materials and Methods section, we provided a clear explanation of the definition of dominant species in Table 1, specifically on page 3, Chapter 2, Materials and Methods, Section 2, Soil Collection and Processing, lines 114 to 115. |
|
Comments 5:In section 2.3, please specify the chemical parameters analyzed and include the names and details of the measurements. Response 5:We sincerely thank you for your careful review and valuable suggestions on this paper. Regarding the suggestion on the description of chemical parameters in Section 2.3, we have given it a careful thought. In the process of writing, we have referred to the expressions in several published literatures in this field, and have moderately streamlined the relevant contents with the requirements of journals for the conciseness of articles. We hope that through this expression, we can ensure scientific rigor and at the same time better fit the journal's writing standards and style. |
|
Comments 6:Indicate which DNA extraction kit was used and provide its full name and manufacturer. Response 6:We sincerely appreciate the opportunity to clarify the methodology employed in our study. For DNA extraction, we utilized the MoBio PowerSoil® DNA Isolation Kit (Catalog No. 12888-100), manufactured by MO BIO Laboratories (now part of QIAGEN). The specific location is on page 4, Chapter 2, Materials and Methods, Section 4, High-throughput sequencing of soil bacteria, first paragraph, lines 131 to 132. |
|
Comments 7:Describe how you processed and normalized the 16S rRNA data, including the specific methods or software used. Response 7:We sincerely appreciate the reviewer's suggestion.We have described in detail how to process and standardise sequencing data. This can be found on page 4, Chapter 2, Section 4, ‘Materials and Methods,’ lines 134 to 162, ‘High-throughput sequencing of soil bacteria.’ |
|
Comments 8:Clarify whether your data met the assumptions required for ANOVA and chi-square tests. Response 8:We sincerely appreciate the reviewer's suggestion. In terms of statistical test assumptions, it is stated that the data passed the Shapiro-Wilk normality test and Levene's test for homogeneity of variance, meeting the prerequisites for ANOVA and chi-square tests. The specific location is on page 5, Chapter 2, Materials and Methods, Section 5, Data Processing and Analysis, lines 164 to 176. |
|
Comments 9:Include a description of PICRUSt2 methodology in the Methods section. Response 9:We sincerely appreciate the reviewer's suggestion. As requested, we have now added a detailed description of PICRUSt2 methodology in the methods section. The specific location is on page 5, Chapter 2, Materials and Methods, Section 4, High-throughput sequencing of soil bacteria, third paragraph, lines 154 to 159. |
|
Comments 10:Adjust the p-values shown in Figure 4, and consider using the FDR or Holm correction method for multiple comparisons. Response 10:Thank you for your inquiry regarding the statistical analysis in Figure 4. For Figure 4 and RDA analysis, the p-values have been adjusted and the FDR-corrected results are marked. The specific location is on page 9, Chapter 3, Section 5, ‘Correlation between Soil Physical and Chemical Properties and Bacterial Community Composition in Different Vegetation Types,’ Figure 4: Comprehensive Analysis of Soil and Microbial Correlation, lines 288 to 291. |
|
Comments 11:Please add and adjust p-values for your RDA analysis to ensure statistical validity. Response 11:We appreciate your suggestion We have re-adjusted the p-value and redrawn Figure 5: RDA analysis of soil physico-chemical properties and bacterial community composition based on phylum level. The updated analysis maintains all previous conclusions while providing enhanced statistical support. The specific location is on page 10, Chapter 3, Section 5, ‘Correlation between Soil Physical and Chemical Properties and Bacterial Community Composition in Different Vegetation Types,’ Figure 5: RDA analysis of soil physico-chemical properties and bacterial community composition based on phylum level, line 301. |
|
Comments 12:I recommend removing Figure 6, as it appears unnecessary. Consider integrating the diversity data into Figure 4 and presenting the results there. Response 12:We sincerely appreciate your valuable comments. Figure 4 mainly shows the relationship between soil physicochemical properties and bacterial phylum community composition, while Figure 6 shows the relationship between soil bacterial community diversity and physicochemical properties under different vegetation types. Combining the graphs would result in too many axes and overlapping information, rather reducing readability. Therefore, we retained the existing presentation for the time being to ensure a clear presentation of the results of both analyses. |
|
Comments 13:For Figure 7, adjust the p-values accordingly. You may also consider removing this figure and incorporating relevant parameters into Figure 4 to avoid redundancy. The current manuscript includes too many figures that are not well justified. Moreover, the discussion lacks depth regarding microbial functional potential and diversity. Response 13:We sincerely thank you for your valuable comments. We have adjusted the p-values of the species in Fig. 7 according to the suggestions. It should be clarified to you here that Fig. 4 focuses on the relationship between soil physicochemical properties and community composition at the bacterial phylum level, while Fig. 7 investigates the relationship between bacterial functional characteristics and soil properties under different vegetation types. As the two figures have different dimensions and focuses of analysis, merging them may lead to redundancy of information and reduce readability. Therefore, we retained the existing split-figure presentation to ensure the clarity of data presentation. The specific location is on page 11, Chapter 3, Results and Analysis, Section 7, Correlation between Soil Bacterial Community Functions and Soil Physico-Chemical Properties in Different Vegetation Types lines 324 to 336. |
|
Comments 14:Expand the conclusion to highlight broader environmental implications, the key questions addressed, the importance of your findings, and limitations (e.g., inferring function based on 16S data). Response 14:We sincerely appreciate the reviewer's suggestion. We have added important findings and limitations to the conclusion, specifically on page 15 of Chapter 5, Conclusion, lines 510 to 542. |
|
Comments 15:The manuscript requires thorough language editing to improve clarity and readability prior to publication. Response 15:Thanks for your suggestion. We have tried ourbest to polish the language in the revisedmanuscript. |

Round 2
Reviewer 1 Report
Comments and Suggestions for Authors
I would like to begin by congratulating the authors for the considerable effort and improvements made to the manuscript. Most of the concerns raised in the previous round have been addressed appropriately. I also acknowledge the inclusion of numerous relevant and recent citations, which significantly strengthen the scientific grounding of the work. However, I kindly ask the authors to replace the current reference no. 24 with the following two updated and more relevant citations: doi.org/10.1016/j.eti.2024.103535 ; doi.org/10.1016/j.scitotenv.2023.166296 These references are more suitable and reflect the current state of knowledge on the topic.